Selection in the dopamine receptor 2 gene: a candidate SNP study

Göllner Tobias tobiasgeht@gmail.com
Fieder Martin
Department of Anthropology, University of Vienna , Vienna , Austria
Erdmann Jeanette
Electronic publication date: 2015 Aug 11
Publication date: 2015
Volume: 3
Electronic Location ID: e1149
Received 2015 May 15; Accepted 2015 Jul 12
Copyright: © 2015 Göllner and Fieder
Copyright year: 2015
Copyright holder: Göllner and Fieder
License: This is an open access article distributed under the terms of the Creative Commons Attribution License, which permits unrestricted use, distribution, reproduction and adaptation in any medium and for any purpose provided that it is properly attributed. For attribution, the original author(s), title, publication source (PeerJ) and either DOI or URL of the article must be cited.
License URL: https://creativecommons.org/licenses/by/4.0/

Keywords: Dopamine, DRD2, Dopamine receptor 2, Schizophrenia, Selection

Funding: Projekt IP547011 Computational Infrastructure was funded by Projekt IP547011 provided by the Faculty of Life Sciences, University of Vienna. The funders had no role in study design, data collection and analysis, decision to publish, or preparation of the manuscript.

==============================
Dopamine is a major neurotransmitter in the human brain and is associated with various diseases. Schizophrenia, for example, is treated by blocking the dopamine receptors type 2. Shaner, Miller & Mintz (2004) stated that schizophrenia was the low fitness variant of a highly variable mental trait. We therefore explore whether the dopamine receptor 2 gene (DRD2) underwent any selection processes. We acquired genotype data of the 1,000 Genomes project (phase I), which contains 1,093 individuals from 14 populations. We included single nucleotide polymorphisms (SNPs) with two minor allele frequencies (MAFs) in the analysis: MAF over 0.05 and over 0.01. This is equivalent to 151 SNPs (MAF > 0.05) and 246 SNPs (MAF > 0.01) for DRD2. We used two different approaches (an outlier approach and a Bayesian approach) to detect loci under selection. The combined results of both approaches yielded nine (MAF > 0.05) and two candidate SNPs (MAF > 0.01), under balancing selection. We also found weak signs for directional selection on DRD2, but in our opinion these were too weak to draw any final conclusions on directional selection in DRD2. All candidates for balancing selection are in the intronic region of the gene and only one (rs12574471) has been mentioned in the literature. Two of our candidate SNPs are located in specific regions of the gene: rs80215768 lies within a promoter flanking region and rs74751335 lies within a transcription factor binding site. We strongly encourage research on our candidate SNPs and their possible effects.

Introduction

The catecholamine dopamine is a neurotransmitter in the human brain. Dopaminergic neurons can be divided into four major pathways: nigrostriatal, mesolimbic, mesocortical and tuberoinfundibular (Andén et al., 1964; Dahlström & Fuxe, 1964). These neurons play an important role in voluntary movement, feeding, reward and learning, as well as certain other functions. Outside the brain, dopamine takes on a physiological role in cardiovascular functions, hormonal regulation, renal and other functions (Snyder et al., 1970; Missale et al., 1998; Sibley, 1999; Carlsson, 2001; Iversen & Iversen, 2007). Due to this involvement in many different processes and systems, dopamine is also related to a variety of diseases. Parkinson’s disease, caused by a loss of dopaminergic innervations in the striatum, is a prominent example (Ehringer & Hornykiewicz, 1960). Additionally, the expected associations between the dopaminergic system and schizophrenia stem from the fact that various dopamine receptor 2 blockers are used as antipsychotics in treating that condition (Snyder et al., 1970; Creese, Burt & Snyder, 1976; Seeman et al., 1976; Carlsson et al., 2001). Further relationships with dopamine dysregulation are expected in Tourette’s syndrome and attention deficit hyperactivity disorder (ADHD) (Mink, 2006; Swanson et al., 2007; Gizer, Ficks & Waldman, 2009). The strong involvement of dopamine in the reward system suggests an association with drug abuse and addiction (Hyman, Malenka & Nestler, 2006; Di Chiara & Bassareo, 2007; Koob & Volkow, 2010). Many more diseases and conditions are expected to involve dopamine dysfunctions. (As reviewed by Beaulieu & Gainetdinov (2011)).

In humans, five different dopamine receptors exist. They are classified into two categories based on their structure and their pharmacological and biochemical properties. The D1-class includes the dopamine receptors 1 and 5, while the D2-class consists of the dopamine receptors 2, 3 and 4 (Andersen et al., 1990; Niznik & Van Tol, 1992; Sibley & Monsma, 1992; Sokoloff et al., 1992; Civelli, Bunzow & Grandy, 1993; Vallone, Picetti & Borrelli, 2000). The focus of our study is on the dopamine receptor 2 and its gene DRD2. The dopamine receptor 2 gene lies on the long arm of chromosome 11 (11q23.1). It spans from 113,280,317 to 113,346,413 for a total of 66,096 base pairs (bp) (information accessed on NCBI in the GnRH37 assembly). For the gene card, see Fig. 1. DRD2 has six introns (Gingrich & Caron, 1993). Alternative splicing between intron 4 and 5 of an 87 bp exon generates two variants of the dopamine receptor 2. The difference between D2S (short) and D2L (long) is a 29-amino-acids-long chain in the third intercellular loop of the protein (Giros et al., 1989; Monsma et al., 1989). While the short form (D2S) is mainly expressed at the presynapse, the long form (D2L) is expressed postsynaptically (Usiello et al., 2000; De Mei et al., 2009). The D2S are mainly autoreceptors, i.e., they reduce the expression of dopamine when activated. This leads to an important negative feedback mechanism (Wolf & Roth, 1990; Missale et al., 1998; Sibley, 1999). (Again, as reviewed by Beaulieu & Gainetdinov, 2011).

Figure 1 Location of candidate SNPs under balancing selection in DRD2.

E1-8 are exons 1 to 8. (1) rs60599314, (2) rs79549222, (3) rs12574471, (4) rs80215768, (5) rs76581995, (6) rs80014933, (7) rs74751335, (8) rs77264605, (9) rs76499333.

Figure 2 Detection of outlier SNPs of the DRD2 gene using FDIST (LOSITAN).

X-axis: estimated heterozygosity values. Y-axis: FST-values. The upper area indicates positive directional selection, the middle area neutrality, and the lower area balancing selection. Confidence intervals represent borders between “selection areas”. See Table S2 for the exact results on the SNPs.

Figure 3 Graphical output of BayeScan.

X-axis: log10(q values) for all SNPs, the threshold is −2. Y-axis: FST-values, where high values indicate directional selection, low values balancing selection. For exact results see Table S2. Squares: the nine SNPs in the MAF > 0.05 sample that BayeScan and LOSITAN find.

Among the many single nucleotide polymorphisms (SNPs) of DRD2, one prominent example is rs6277, also known as C957T. It has been associated with schizophrenia in Han Chinese in Taiwan (Glatt et al., 2009), in Russians (Monakhov et al., 2008) and in Bulgarians (Betcheva et al., 2009). Together with the -141C allele, the 957T allele is associated with the diagnosis of anorexia nervosa (Bergen et al., 2005). A meta-analysis showed that the Ser311Cys polymorphism (rs1801028) in DRD2 is a risk factor for schizophrenia. The heterozygotes (Ser/Cys) and the homozygotes for Cys were both at elevated risk for schizophrenia when compared to the Ser/Ser genotypes (Glatt & Jönsson, 2006). In a study with alcoholic patients and controls, the A allele of rs1076560 was more frequent in alcoholic patients (Sasabe et al., 2007). In 2012, Mileva-Seitz et al. conducted a study with Caucasian mothers and their infants. They taped mother-infant behaviour and genotyped various SNPs of DRD2 and also DRD1. rs1799732 and the previously mentioned rs6277 were both associated with direct vocalization of the mother towards the infant.

The body of literature on SNPs and their possible effects is growing rapidly. Considering the influences those SNPs could have on human behaviour, and bearing in mind the different ecological habitats of Homo sapiens, we explore if DRD2 underwent any selection processes. In 2004, an interesting proposal by Shaner, Miller & Mintz stated that schizophrenia was the low fitness variant of a highly variable mental trait. Based on the connection between dopamine receptor 2 and schizophrenia, as stated above, we focused our analysis on DRD2.

To reduce false-positives, we used two selection detection algorithms to explore DRD2. This exploratory (“hypothesis-free”) approach is designed to find candidate SNPs that were under selection. The data basis of our analysis is the 1,000 Genomes Project samples.

Material and Methods

We acquired data from the 1,000 Genomes Project (phase I) through SPSmart engine v5.1.1 (http://spsmart.cesga.es/engines.php; Amigo et al., 2008), using the search term “DRD2.” We included all single nucleotide polymorphisms (SNPs) with a minor allele frequency (MAF) greater than 0.05 (N = 151 SNPs) to include only the more frequently occurring SNPs. To verify our results also on the basis of a higher number of SNPs (which occur less frequently), we conducted the same analysis also based on a MAF > 0.01 sample (N = 246 SNPs; data presented in Supplemental Information). The structure of the DRD2 gene (113,280,317–113,346,413 in the GnRH37.p13 primary assembly) is shown in Fig. 1. The populations used for our analysis are shown in Table 1.

The data were converted by hand into the CONVERT format. All further format conversions were performed by PGD Spider 2.0.5.2 (Lischer & Excoffier, 2012).

Two different programs were used to detect selection; both use FST approaches to detect outliers. The program LOSITAN calculates FDIST, which uses FST and the expected heterozygosity. It assumes an island model of migration with neutral markers. An expected distribution of Wright’s inbreeding coefficient is calculated and then outliers are identified. A neutral mean FST was computed by the program before the 50,000 simulations were performed. The infinite alleles model was used. To avoid false positive detection, we set the significance level to p < 0.01 (P(Simulation FST < sample FST)) (Antao et al., 2008).

BayeScan is a Bayesian statistics program. Basically, it calculates two simulations for every locus: one in which it assumes the locus is under selection and the other one in which this assumption is dropped. It splits the FST coefficient into two parts. The alpha value is a locus-specific component shared by all populations. The beta value is a population-specific component shared by all loci. This is achieved via logistic regression and provides insight into selection. The alpha value serves as an indicator for selection. Significant positive values of alpha indicate directional selection, whereas significant negative values indicate balancing selection. The posterior probabilities are estimated using a reversible-jump Markov Chain Monte Carlo (MCMC) approach. The posterior probabilities are gained by counting how many times alpha is included in the model. Before computing the Markov chains, we calculate 20 pilot runs with 5,000 iterations each. The initial burn-in is set to 50,000 steps and the chains are run with 5,000 iterations and a thinning interval of 10. The program output consists of a posterior probability, the logarithm (base 10) of the posterior odds and a q value. These three values are all for the model with selection. Furthermore, the alpha value is reported along with an FST coefficient average of all population per locus. In BayeScan the threshold of a posterior P of >0.99 and a log10(PO) of 2 or higher is used. This threshold is labelled as “Decisive” by BayeScan (see the program manual at http://cmpg.unibe.ch/software/BayeScan/files/BayeScan2.1_manual.pdf) (Foll & Gaggiotti, 2008).

To compute linkage disequilibrium (LD) of the SNPs, we used the R “genetics package” (http://cran.r-project.org/web/packages/genetics/genetics.pdf; Warnes et al., 2013). Mueller (2004) states that D′ is particularly useful to assess the probability for historical recombination in a given population and r2 is useful in the context of association studies. We therefore primarily calculate D′, but we also calculated r2, which is presented in Supplemental Information.

In most populations one or more SNPs had to be excluded to successfully run the computation. The population IBS was excluded entirely from this computation. IBS is a very small population (n = 14), and 30 SNPs caused the computation to fail. For a detailed view on all excluded SNPs, see Table S1.

We accessed information on the gene via NCBI (http://www.ncbi.nlm.nih.gov/) and on the specific SNPs via Ensembl (http://www.ensembl.org/).

Results

The combined results of LOSITAN and BayeScan yielded nine candidate SNPs under balancing selection (MAF > 0.05); see Table 2. Figure 2 shows the graphical output of LOSITAN and Fig. 3 the output of BayeScan.

For a detailed view on the results of LOSITAN and BayeScan for all SNPs, see Table S2 in the supplementary material. Figure 1 provides a gene view of DRD2 with labels for the candidate SNPs.

The same calculations based on the MAF > 0.01 sample revealed only 2 SNPs (rs60599314, rs79549222) under balancing selection by both LOSITAN and Bayescan (Fig. S1, Table S3A and Table S3B).

Three SNPs (rs6277, rs12800853, rs11608109) that do not reach significance in the MAF > 0.05 sample (Table S3A) do reach significance in the MAF > 0.01 sample (Table S3B), for directional selection. They barely reach significance (P < 0.01) based on the MAF > 0.01 sample in LOSITAN (Fig. S1 and Table S3B). However if we applied more stringent detection prerequisites (“force mean FST” and “neutral mean FST”; increasing computational load, but also increasing convergence and lowering the bias in FST estimation) in LOSITAN, none of these three SNPs reaches significance (Fig. S2). The results for balancing selection in BayeScan remained nearly unchanged in the MAF > 0.01 sample, with the exception of rs12574471, which did not reach significance (Table S3B).

All nine SNPs detected based on MAF > 0.05 are intron variants (Fig. 1). Only rs12574471 (3) is mentioned in the literature because it is near a supposed recombination hotspot (Glatt et al., 2009). rs80215768 (4) lies within a promoter flanking region; rs74751335 (7) lies within a transcription factor binding site. Nonetheless, we found no known associations for those two SNPs.

The FST values of these nine loci indicate an overall low genetic differentiation, as well as a low differentiation between populations (Table 2). This is in accordance with balancing selection acting on the gene. The differences in FST values stem from different algorithms used by the programs.

Table 1 Populations of the 1,000 genomes project.

Superpopulation (code)	Population code	Population	Number of individuals	
Africa (AFR)	ASW	African ancestry in Southwest USA	61	
LWK	Luhya in Webuye, Kenya	97	
YRI	Yoruba in Ibadan, Nigeria	88	
Europe (EUR)	CEU	Utah residents with Northern and Western European ancestry	87	
FIN	Finnish from Finland	93	
GBR	British from England and Scotland	88	
IBS	Iberian populations in Spain	14	
TSI	Toscani in Italy	98	
East Asia (ASN)	CHB	Han Chinese in Bejing, China	97	
CHS	Han Chinese South	100	
JPT	Japanese in Tokyo, Japan	89	
America (AMR)	CLM	Colombians from Medellin, Colombia	66	
MXL	Mexican ancestry from Los Angeles USA	60	
PUR	Puerto Ricans from Puerto Rico	55	
		All populations	1,093	

Table 2 The dopamine receptor 2 gene’s nine candidate SNPs for balancing selection (MAF > 0.05).

Locus (#)	Major allele (Frequency)	Minor allele (Frequency)	FST (Lositan)	FST (BayeScan)	Location	
rs60599314 (1)	C (0.871)	T (0.129)	0.0110	0.0272	113,306,431 (Intronic region)	
rs79549222 (2)	T (0.87)	G (0.13)	0.0106	0.0260	113,310,340 (Intronic region)	
rs12574471 (3)	C (0.891)	T (0.109)	0.0172	0.0364	113,316,236 (Intronic region)	
rs80215768 (4)	G (0.925)	A (0.075)	0.0304	0.0328	113,318,880 (Intronic region)	
rs76581995 (5)	C (0.925)	A (0.075)	0.0304	0.0328	113,319,835 (Intronic region)	
rs80014933 (6)	T (0.923)	C (0.077)	0.0304	0.0332	113,328,135 (Intronic region)	
rs74751335 (7)	G (0.915)	C (0.085)	0.0266	0.0322	113,328,810 (Intronic region)	
rs77264605 (8)	A (0.915)	G (0.085)	0.0266	0.0327	113,328,913 (Intronic region)	
rs76499333 (9)	G (0.925)	A (0.075)	0.0299	0.0327	113,329,449 (Intronic region)	

The Linkage Disequilibrium measurements D′ and r2 were used. The heat maps for all nine populations are shown in the supplementary material (Figs. S3–S15 for D′ and Figs. S16–S28 for r2). The relative position of the marked SNPs change because different populations had different SNPs excluded (see Table S1 for the list).

Discussion

We found nine SNPs to be candidates for balancing selection based on the sample MAF > 0.05; of those, two had been also detected under balancing selection based on the MAF > 0.01 sample. We found no SNPs based on the MAF > 0.05 sample and the MAF > 0.01 sample, under directional selection, that are detected by both algorithms (on P < 0.001 in LOSITAN) and if more stringent detection criteria were applied in LOSITAN. We therefore conclude that, if directional selection has been acting on DRD2, then the signs are rather weak, i.e., too weak to make definitive conclusions.

Checking all nine SNPs under balancing selection based on the MAF > 0.05 sample with Ensembl reveals that they are all intronic region variants. For rs60599314 (1) and rs79549222 (2) that are found by LOSITAN and BayeScan on both the MAF >0.05 sample and the MAF >0.01 sample, no particular additional information is known. We therefore suggest that these two SNPs may provide interesting candidates for future functional studies.

rs80215768 (4) lies within a promoter flanking region and rs74751335 (7) lies within a transcription factor binding site (TFBS) (both SNPs detected based on the MAF >0.05 sample). Many studies are available on the possible effects of mutations in such regions (Hayashi, Watanabe & Kawajiri, 1991; In et al., 1997; or for a more general review on the topic, Jaenisch & Bird, 2003). Nonetheless, the SNPs show low FST values, which is congruent with the finding of balancing selection. Sewall Wright’s guidelines for interpreting FST values suggest little genetic differentiation in our populations (as cited by Jobling et al., 2013; Chapter 5, Box 5.2). As silent mutations in DRD2 are known to alter the mRNA stability and even the synthesis of the receptor itself (Duan et al., 2003), we call for exploring the possible effects of these SNPs.

Additionally, the levels of the linkage disequilibrium measurement D′ are typical for the respective populations: African populations show a dispersed pattern and no clear LD blocks (Figs. S3, S11, and S15). While the LD blocks are visible in American populations (Figs. S7, S12 and S13), they are not as clear as in Asian (Figs. S5, S6 and S10) or European populations (Figs. S4, S8, S9 and S14). Our candidate SNPs are part of tight LD blocks (D′ > 0.8), which prevents us from making any further interpretations. We also examined the measurement r2 for all populations (again, excluded SNPs are listed in Table S1), which revealed no new insight.

The finding of balancing selection suggests that in our sample the minor alleles bear some fitness disadvantage. Note that some individuals are homozygous for the minor allele (0.8–2.3% per SNP, over all populations). Fitness is altered if survival or reproduction of an organism is affected. This raises the possibility of a connection between our candidate SNPs and diseases or malfunctions of dopamine receptor 2. In the list of diseases associated with dopamine (see “Introduction”) the most striking example is schizophrenia because dopamine receptor 2 blockers can successfully treat patients.

Albeit we aim to avoid overhasty conclusions regarding directional selection on DRD2, the three SNPs detected by BayeScan under positive selection—and that are found by LOSITAN just below significance—could be of interest. This is particularly the case for rs6277, with its known phenotypic associations: rs6277 has been associated with schizophrenia in Han Chinese in Taiwan (Glatt et al., 2009), in Russians (Monakhov et al., 2008) and in Bulgarians (Betcheva et al., 2009).

Nonetheless, rs6277 was not identified among the 108 schizophrenia-associated loci that have recently been published based on 36,989 cases and 113,075 controls by the Schizophrenia Working Group of the Psychiatry Genomics Consortium (Schizophrenia Working Group of the Psychiatric Genomics Consortium, 2014). Nevertheless, DRD2 is one of the genes that was confirmed as being associated with schizophrenia by the Working Group. Thus, even weak signs of directional selection on DRD2 might be important to (i) identify potential new disease-related phenotypical associations and (ii) to speculate on what the “selective force” could have been bringing mutations on DRD2 towards fixations and (iii) what potential condition-related consequences selection on DRD2 could have. The question is whether these conditions affect fitness. Accordingly, Bassett et al. (1996) showed that reproductive fitness is reduced in groups of familial schizophrenia, which suggests a selection process. Puzzlingly enough, they also found some evidence for an increased fitness of a small subsample of sisters. Shaner, Miller & Mintz (2004) proposed that schizophrenia is the low-fitness trait of a highly variable mental trait. They argue that the persistence of the illness at about 1% globally is too high for new mutations. Thus, mainly balancing selection would fit this hypothesis very well, and our candidate SNPs under balancing selection could be viable indicators for this.

DRD2 is clearly associated with schizophrenia. Schizophrenia, however, is a “polygenic condition” including genetic loci over the whole human genome (Schizophrenia Working Group of the Psychiatric Genomics Consortium, 2014). Accordingly, the importance of DRD2 should not be over-estimated. Moreover, the method of selection detection does not allow direct inferences about a phenotype (e.g., schizophrenia). Our overall results can serve as a valuable precursor to future studies on the subject.

To untangle the possible effects of our SNPs, we propose a study in which our candidate SNPs are investigated in schizophrenic and non-schizophrenic persons. A simple comparison of the SNPs and the different haplotypes between the two groups should efficiently help assess our findings. If this proposed study finds differences in those two groups, then the mechanisms of those SNPs and their possible haplotypes must be investigated.

Conclusion

We found nine candidates for balancing selection on DRD2 but only a weak signs for directional selection. Interestingly, rs6277, a SNP known to be associated with schizophrenia, is among those SNPs for which we found weak evidence for directional selection. Some of the SNPs under balancing selection are potentially associated with various diseases. These SNPs could be important as biomarkers due to their very low FST values: the genetic differentiation of one population compared with the whole sample is very small. While all candidate SNPs may be worth exploring, we definitely recommend using rs60599314 and rs79549222 (as these were detected under balancing selection based on a MAF >0.05 and a MAF >0.01 sample). We also recommend rs80215768 and rs74751335, found under balancing selection, for further studies on DRD2 because the former within a promoter flanking region and the latter lies in a transcription factor binding site.

Supplemental Information

Figure S1 LOSITAN output for the MAF > 0.01 sample

X-axis: estimated heterozygosity values. Y-axis: FST-values. The red area indicates positive directional selection, the grey area indicates neutrality, and the yellow area indicates balancing selection. Confidence intervals represent borders between “selection areas”.

Click here for additional data file.

Figure S2 LOSITAN output for the MAF > 0.01 sample, with more stringent criteria applied (“force mean FST” and “neutral mean FST”)

X-axis: estimated heterozygosity values. Y-axis: FST-values. The red area indicates positive directional selection, the grey area indicates neutrality, and the yellow area indicates balancing selection. Confidence intervals represent borders between “selection areas”.

Click here for additional data file.

Figure S3 D′ heat map of the ASW population

Click here for additional data file.

Figure S4 D′ heat map of the CEU population

Click here for additional data file.

Figure S5 D′ heat map of the CHB population

Click here for additional data file.

Figure S6 D′ heat map of the CHS population

Click here for additional data file.

Figure S7 D′ heat map of the CLM population

Click here for additional data file.

Figure S8 D′ heat map of the FIN population

Click here for additional data file.

Figure S9 D′ heat map of the GBR population

Click here for additional data file.

Figure S10 D′ heat map of the JPT population

Click here for additional data file.

Figure S11 D′ heat map of the LWK population

Click here for additional data file.

Figure S12 D′ heat map of the MXL population

Click here for additional data file.

Figure S13 D′ heat map of the PUR population

Click here for additional data file.

Figure S14 D′ heat map of the TSI population

Click here for additional data file.

Figure S15 D′ heat map of the YRI population

Click here for additional data file.

Figure S16 r2 heat map of the ASW population

Click here for additional data file.

Figure S17 r2 heat map of the CEU population

Click here for additional data file.

Figure S18 r2 heat map of the CHB population

Click here for additional data file.

Figure S19 r2 heat map of the CHS population

Click here for additional data file.

Figure S20 r2 heat map of the CLM population

Click here for additional data file.

Figure S21 r2 heat map of the FIN population

Click here for additional data file.

Figure S22 r2 heat map of the GBR population

Click here for additional data file.

Figure S23 r2 heat map of the JPT population

Click here for additional data file.

Figure S24 r2 heat map of the LWK population

Click here for additional data file.

Figure S25 r2 heat map of the MXL population

Click here for additional data file.

Figure S26 r2 heat map of the PUR population

Click here for additional data file.

Figure S27 r2 heat map of the TSI population

Click here for additional data file.

Figure S28 r2 heat map of the YRI population

Click here for additional data file.

Table S1 SNPs that caused the LD calculations of R to fail

These SNPs were excluded from the LD calculations. The population IBS was excluded entirely because 30 SNPs were flawed (which is nearly 20% of all SNPs).

Click here for additional data file.

Table S2 Results of LOSITAN (left) and BayeScan (right) for all 151 SNPs of DRD2 (MAF > 0.05)

P, P (Simulation FST < sampleFST); He, expected heterozygosity; FST, Fixation Indices subpopulation to total population. P, posterior probability; log10 (PO), logarithm (base 10) of the posterior odds; q-value, false discovery rate (FDR) analogue of the P value; αi = locus-specific component (negative alpha suggests balancing selection, while positive alpha suggests directional selection); FST, Fixation Indices subpopulation to total population. ∗—the value of 1,000 means infinity (see BayeScan manual).

Click here for additional data file.

Table S3A Candidate loci under selection in the MAF > 0.05 sample, plus the candidates of the MAF > 0.01 sample

Comparison of the candidate loci under selection found both samples; (a) shows the calculations of the MAF > 0.05 sample and the (b) the calculations of the MAF > 0.01 sample. P, P (Simulation FST < sample FST); He, expected heterozygosity; FST, Fixation Indices subpopulation to total population. P, posterior probability; log10 (PO), logarithm (base 10) of the posterior odds; q-value, false discovery rate (FDR) analogue of the P value; αi, locus-specific component (negative alpha suggests balancing selection, while positive alpha suggests directional selection); FST, Fixation Indices subpopulation to total population.

Click here for additional data file.

Table S3B Candidate loci under selection in the MAF > 0.01 sample, plus the candidates of the MAF > 0.05 sample

Comparison of the candidate loci under selection found both samples; (a) shows the calculations of the MAF > 0.05 sample and the (b) the calculations of the MAF > 0.01 sample. P P (Simulation FST < sample FST); He, expected heterozygosity; FST, Fixation Indices subpopulation to total population. P, posterior probability; log10 (PO), logarithm (base 10) of the posterior odds; q-value, false discovery rate (FDR) analogue of the P value; αi, locus-specific component (negative alpha suggests balancing selection, while positive alpha suggests directional selection); FST, Fixation Indices subpopulation to total population.

Click here for additional data file.

Supplemental Information 33 CONVERT datafile of the MAF > 0.05 sample

Click here for additional data file.

Supplemental Information 34 CONVERT datafile of the MAF > 0.01 sample

Click here for additional data file.

The authors would like to thank the R Core Team for the statistical computing environment, the 1,000 Genomes Project, as well as the creators of LOSITAN, BAYESCAN and PGD Spider. The University of Vienna for the “Investitionsprojekt IP 547011.” Additional thanks go to Bernard Wallner, Philipp Gewessler, Matthias Hirschmanner, Lukas Schwabegger and Michael Stachowitsch. Furthermore we thank the reviewers for their valuable comments that helped to improve the manuscript.

Additional Information and Declarations

Competing Interests

Author Contributions

The authors declare that there are competing interests.

Tobias Göllner conceived and designed the experiments, performed the experiments, analyzed the data, contributed reagents/materials/analysis tools, wrote the paper, prepared figures and/or tables.

Martin Fieder conceived and designed the experiments, contributed reagents/materials/analysis tools, reviewed drafts of the paper.

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
