# Peer review of "Selection in the dopamine receptor 2 gene: a candidate SNP study"

_PeerJ, doi:10.7717/peerj.1149_

## Round 0.1 · original submission · Major Revisions

Dear Dr. Göllner,

Please answer specifically to the comments of Reviewer 2.

·

Basic reporting

Figure 2: A little more explanation of the output would help. Grey=light blue? Is there a solution for the overprinting of SNP numbers like symbols or colours?
Figure 3: The representation of SNPs as numbers is not so decent. Maybe symbols would give a clearer graphic.

Experimental design

Why is there a threshold of 5% MAF?
Why was D' taken as Linkage Disequilibrium measurement? D' takes not into account if SNPs have different MAFs.

Validity of the findings

No Comments

Reviewer 2 ·

Basic reporting

Acceptable.

Experimental design

The experimental design fits to the PeerJ standards within the scope of the journal. However, I am not adequately qualified to comment in detail on the statistical methods used on the data.

Validity of the findings

Acceptable.

Additional comments

This manuscript reports on an investigation of selection in the dopamine receptor 2 gene. Please find two comments below:
1. Whilst this reviewer is not an expert in neurology or psychiatry, it has to be asked whether there are any links between directional selection and schizophrenia. Maybe the authors could comment on this question.
2. In 2014, the Schizophrenia Working Group of the Psychiatric Genomics Consortium (Nature, 2014; doi:10.1038/nature13595) reported 108 loci to be genome-wide significantly associated with schizophrenia. Being DRD2 one of these loci, the SNPs detected in this study have not been found to be associated. The findings in this study should be discussed in the light of association studies.
3. The manuscript mainly presents data on selection in the DRD2 gene by a bioiformatical approach. The conclusions of the paper should be limited to this finding which ist still interesting.

---

## Round 0.2 · accepted · Accept

No further comments - the comments made by the reviewers have been fully adressed.